# Visual processing of informative multipoint correlations arises primarily in V2

**Yunguo Yu, Anita M Schmid, Jonathan D Victor\***

Brain and Mind Research Institute, Weill Cornell Medical College, New York, United States

**Abstract** Using the visual system as a model, we recently showed that the efficient coding principle accounted for the allocation of computational resources in central sensory processing: when sampling an image is the main limitation, resources are devoted to compute the statistical features that are the most variable, and therefore the most informative (eLife 2014;3:e03722. DOI: 10.7554/eLife.03722 *Hermundstad et al., 2014*). Building on these results, we use single-unit recordings in the macaque monkey to determine where these computations—sensitivity to specific multipoint correlations—occur. We find that these computations take place in visual area V2, primarily in its supragranular layers. The demonstration that V2 neurons are sensitive to the multipoint correlations that are informative about natural images provides a common computational underpinning for diverse but well-recognized aspects of neural processing in V2, including its sensitivity to corners, junctions, illusory contours, figure/ground, and 'naturalness.'

## Introduction

We recently showed (eLife 2014;3:e03722. DOI: 10.7554/eLife.03722 [*Hermundstad et al., 2014*]) how a normative theory based on the efficient coding principle (*Barlow, 1961*) can account for the allocation of resources for the representation of complex sensory features. Specifically, we analyzed the local statistics of natural images, and compared the variability of these statistics with their perceptual salience. The statistics that were the most variable—that is, the least predictable and therefore the most informative—were the most salient perceptually. This relationship, in which greater resources are allocated to more variable features, emerges from the efficient coding principle in the regime that the main constraint is input sampling (*Barlow, 1961*; *van Hateren, 1992*; *Doi and Lewicki, 2014*; *Hermundstad et al., 2014*). The observed relationship contrasts with the more familiar 'whitening' regime (*Srinivasan et al., 1982*), which emerges when the main constraint is output capacity (e.g., with regard to the retina and the optic nerve bottleneck); the whitening regime predicts that fewer resources are allocated to more variable features. We note that the results of (*Hermundstad et al., 2014*) provide empirical support for the hypothesis that input sampling, rather than output capacity, is the main constraint—since a transmission limit would have predicted a lower sensitivity for image statistics that were the most variable, the opposite of what we found.

To reach this result, we analyzed natural images via their multipoint correlations, that is, the statistics of the combinations of luminance values that appear in several points of the image. This approach has several advantages. First, it reduces the dimensionality of the space of image statistics that need to be considered, since it can be applied to binarized images, and it separates informative from uninformative statistics (*Tkačik et al., 2010*). Second, the approach enables rigorous tests of theoretical predictions, since the individual kinds of informative and uninformative multipoint correlations can be isolated in synthetic image sets (*Victor and Conte, 2012*). In contrast, the multipoint correlations in natural images

**\*For correspondence:** jdvicto@ med.cornell.edu

**Competing interests:** The authors declare that no competing interests exist.

covary in a complex manner that is difficult to capture or control. Synthetic image sets that isolate individual kinds of multipoint correlations are highly un-natural, but here this is an advantage: our predictions, which are derived from natural images, are tested in an out-of-sample fashion.

The information-theoretic framework of (*Hermundstad et al., 2014*) and (*Tkačik et al., 2010*) played a key role in this analysis, and we briefly summarize it here. We used a two-stage model: first, the informative multipoint features (as identified by [*Tkačik et al., 2010*]) are extracted from a visual image by a set of local nonlinear processing elements. Then, the output of this stage, that is, the frequency of each feature in patches of the image, is represented and transmitted by central visual areas, to serve as the basis for visual inferences (Figure 4C of [*Hermundstad et al., 2014*]). We used a linear channel with additive Gaussian noise as an approximation for this latter process. While obviously a simplification, this leads to an analytic solution (*van Hateren, 1992*) for the allocation of resources that maximizes the mutual information between stimuli and their central representation—and the analytic solution accurately accounted for dozens of independently-determined psychophysical parameters (*Hermundstad et al., 2014*).

Unaddressed, however, was where the extraction of multipoint correlations takes place. Several lines of evidence suggested that selective sensitivity to multipoint correlations arises in visual cortex (discussed in [*Hermundstad et al., 2014*]), but a direct demonstration was lacking.

Here, we report single-unit recordings in macaque V1 and V2, showing that neuronal selectivity for multipoint correlations is infrequent in V1, and becomes prominent in V2, especially in its supragranular layers. Well-recognized characteristics of V2 neurons, including sensitivity to corners, junctions (*Das and Gilbert, 1999*), illusory contours (*von der Heydt et al., 1984*), figure/ground (*Qiu and von der Heydt, 2005*), and 'naturalness' (*Freeman et al., 2013*) all entail sensitivity to multipoint correlations; here we show that this sensitivity is present even when these correlations are separated from their natural context.

## Results

We recorded the extracellular activity of 421 individual neurons (269 in V1, 152 in V2) in the anesthetized, paralyzed macaque to stimulus sets that isolate the multipoint correlations previously studied in natural images (*Tkačik et al., 2010*; *Hermundstad et al., 2014*) and psychophysically (*Victor and Conte, 1991*, *2012*).

The stimulus sets are illustrated in the top row of *Figure 1* (see 'Materials and methods' for details). In the '*random*' stimulus set, check colors are assigned independently, with an equal chance of being white or black. The six structured stimulus sets were as follows: The '*even*' and '*odd*' sets isolate the opposite extremes of the visually salient four-point correlation (*Hermundstad et al., 2014*), there denoted $\alpha$. The '*white triangle*' and '*black triangle*' sets isolate the extremes of the visually salient three-point correlation (*Hermundstad et al., 2014*), there denoted $\theta$. The '*wye*' and '*foot*' sets have multipoint correlations are not visually salient (*Victor and Conte, 1991*); this is in keeping with the efficient coding principle because in natural images, these correlations are predictable from simpler quantities (*Tkačik et al., 2010*). We focused on three- and four-point correlations, since one- and two-point statistics (luminance and spatial contrast) are well-known to modulate responses throughout the visual system, beginning in the retina.

*Figure 1* shows post-stimulus histograms (PSTHs) of typical neurons in V1 and V2. Responses have a prominent transient after each stimulus transition, when on average half of the checks change from black to white or from white to black. For some neurons (e.g., the first, third, fourth, and sixth examples in V1), this transient is nearly identical for each of the stimulus sets. For other neurons (e.g., the second and fifth examples in V1, and most of the V2 examples), the transients differ in magnitude or configuration, suggesting a differential response to multipoint correlations.

To quantify these differences, we applied a shuffle test to the smoothed firing rates (see 'Materials and methods'). Significant differences between responses to structured and random stimuli (the asterisks in *Figure 1*) were more common in V2 than in V1. For a more thorough characterization, we defined the 'multipoint correlation discrimination index' (MCDI), which counted not only the comparison between structured and random stimuli, but also comparisons among pairs of different structured stimuli. The MCDI was defined as the fraction of the 21 pairwise comparisons that differed by the above statistical criterion. An MCDI of 0 means that a neuronal response to all stimulus types is indistinguishable; an MCDI of 1 means that a neuronal response distinguishes between all stimulus pairings, and therefore, between all of the structured stimulus sets.

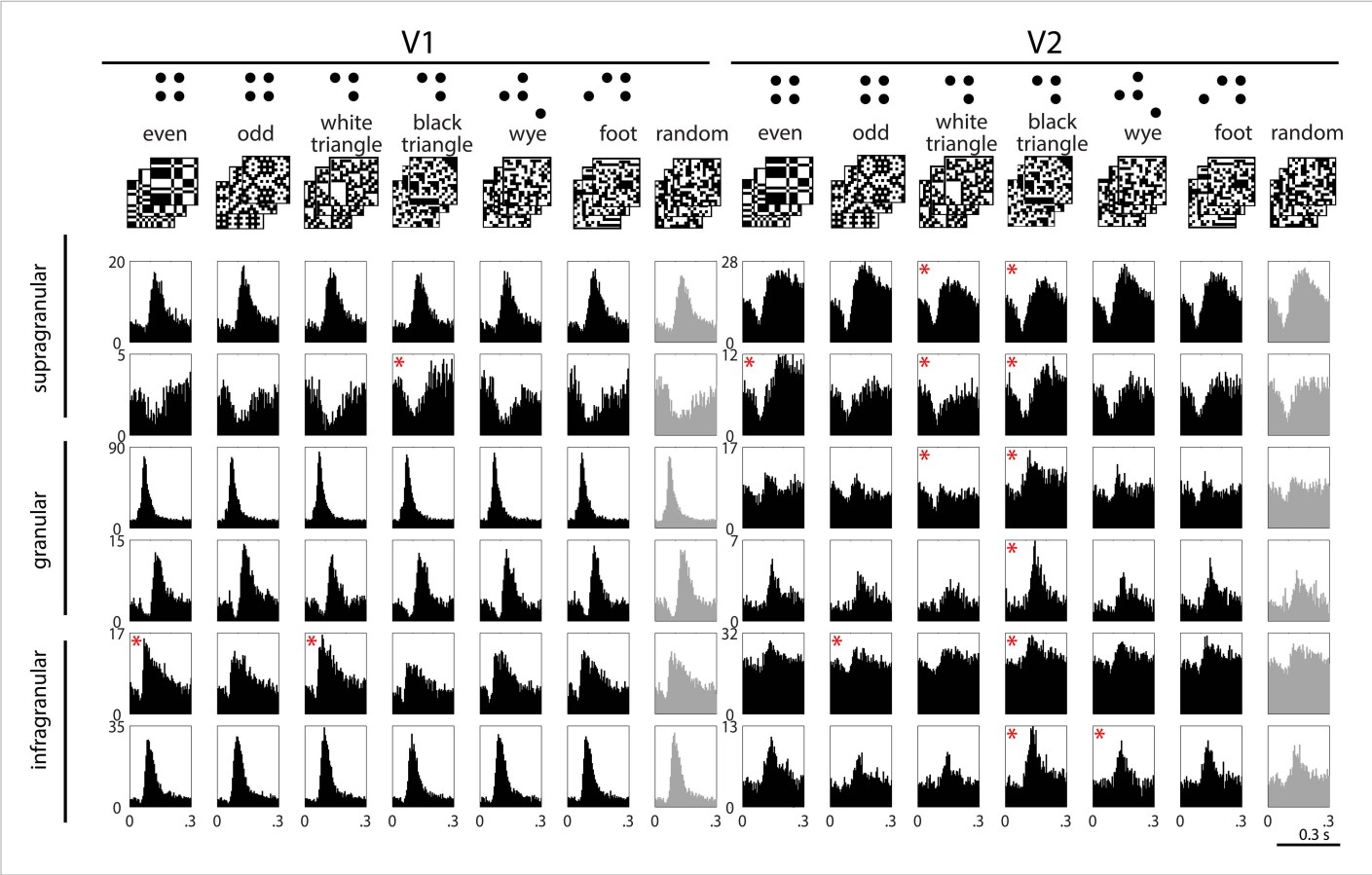

**Figure 1**. Example responses to multipoint correlations in V1 and V2. Top row: examples of the stimulus sets used to isolate the different kinds of multipoint correlations. Six sets consist of 1024 16 × 16 binary checkerboards, each with a different statistical structure (left columns); the seventh set consists of 1024 16 × 16 random checkerboards (right column); see 'Materials and methods' for details. In each column, the row of PSTH's shows responses of a single neuron to 1024 examples of stimuli drawn from the seven sets. Responses are generally dominated by a transient increase or decrease in firing, occurring 70 to 100 ms after the onset of each stimulus. In some cases, the size or configuration of this transient depends on the type of multipoint correlation (for example, the units in the second row). The asterisks indicate responses to the structured stimulus sets (black) that are significantly different (see 'Materials and methods') from the responses to the random stimuli (light gray, beginning of each row). Decremental responses following contrast onset were present in both areas, but more often in V2. However, a decremental response was not a requirement for discriminating among multipoint correlations: outside of supragranular V2, there were many neurons that had incremental responses to the stimulus transient and distinguished among the types of multipoint correlation (for example, the third and fourth rows on the right).

*Figure 2A* (upper row) summarizes the MCDI across the neuronal population. The average MCDI peaked at a value of approximately 0.05 in V1, and approximately 0.10 in V2; this difference became significant at 70 ms after stimulus onset.

A laminar analysis of the MCDI (*Figure 2A*, lower three rows) revealed a slight increase from the V1 granular (input) layer (mean 0.025) to the V1 extragranular layers (supragranular: 0.033, infragranular, 0.045), followed by a jump at the V2 granular layer (0.101), with a marked upsurge in the V2 supragranular layer (0.162), but not the infragranular layer (0.052). The difference between the MCDI in supragranular V2 and each of the other compartments was significant, except for the comparison with granular V2 (p = 0.053). The median value of the MCDI in supragranular V2 was 0.12, meaning that the 'typical' neuron responded differentially for 2 or 3 of the 21 pairwise comparisons. In all other compartments (in V1 and V2), the median was 0, that is, the 'typical' neuron did not distinguish between any of the stimulus types. Atypical neurons in V1 did distinguish among multipoint correlations. These were primarily neurons in the infragranular layer and with large receptive fields (RFs)—see *Figure 2—figure supplement 1*. But overall, the mean MCDI was lower in V1 (0.027) than in V2 (0.081), especially in its supragranular compartment (0.162).

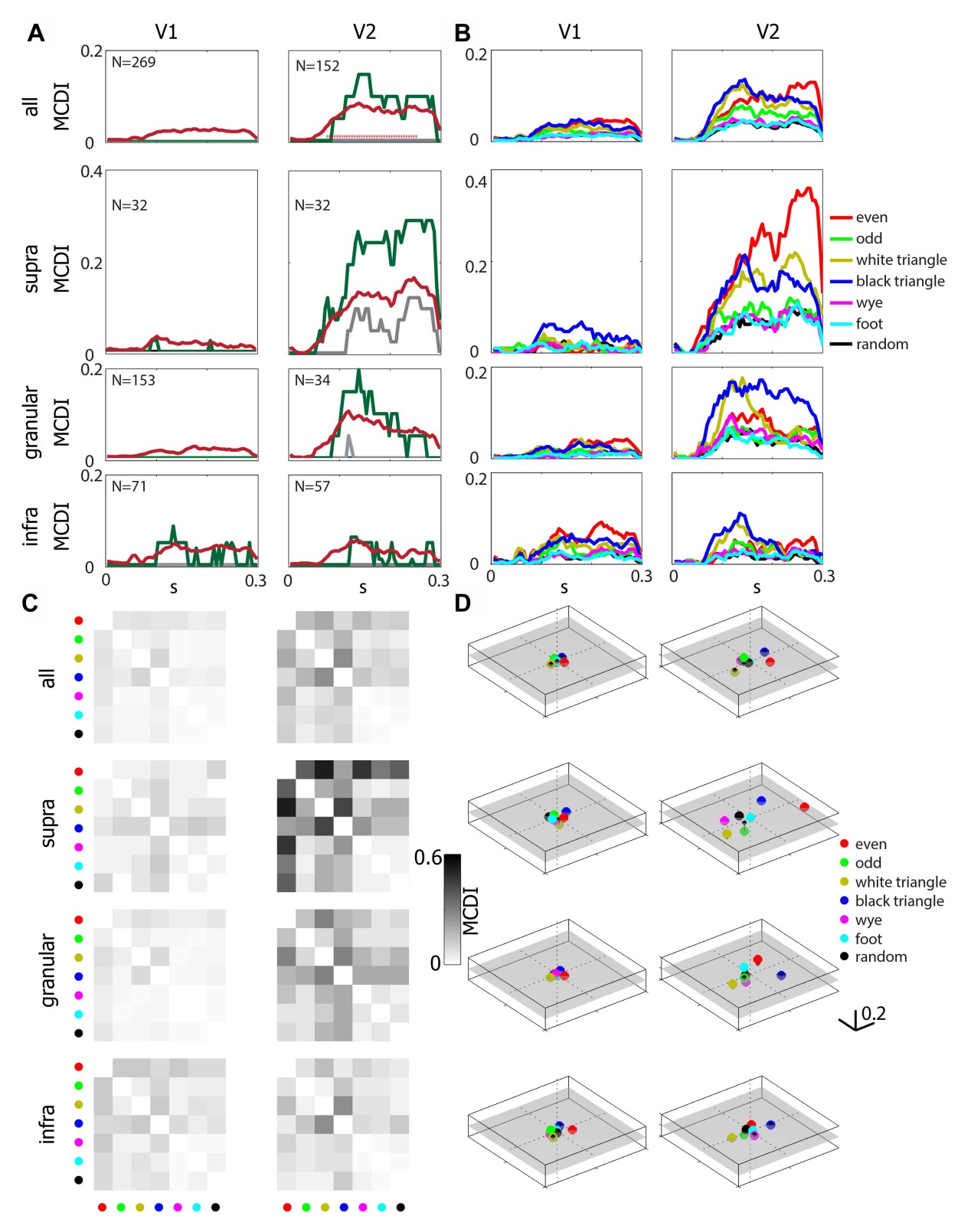

**Figure 2**. Differential sensitivity to multipoint correlations arises intracortically, primarily in V2, and are selective for informative (*Tkačik et al., 2010*) multipoint correlations. (**A**) The multipoint correlation discrimination index (MCDI, see 'Materials and methods') for all stimulus types. Upper panels include all neurons in each area, lower three rows subdivide according to lamina. Mean (dark red), median (gray), and 75th percentile (dark green).

*Figure 2. continued on next page*

*Figure 2. Continued*

25th percentile is 0 in all cases. The red dots in the upper right panel indicate a significant difference between V2 and V1 (p < 0.05, two-tailed, Wilcoxon rank-sum test, false-discovery-rate corrected). The number in the upper left of each panel indicates the number of units analyzed. (**B**) Mean values of the stimulus-specific MCDI. The stimuli with the highest contributions are the ones that contain correlations that are informative for natural images (*Tkačik et al., 2010*): *even* (red), *odd* (green), *white triangles* (yellow) *black triangles* (blue). In contrast, the others (*random* (black), *wye* (magenta), and *foot* (cyan)) are uninformative for natural images, and contributed little to the MCDI. (**C**) Pairwise discrimination of the multipoint correlation types. The grayscale shows the average pair-specific MCDI, which is the fraction of neurons that respond differentially at any time from 55 to 250 ms following stimulus onset. The stimuli for each row and column are indicated by the same color code as in panel **B**. Note that panel **A** shows the overall MCDI, panel **B** shows the stimulus-specific MCDI, and panel **C** shows the pair-specific MCDI. (**D**) Multidimensional scaling of the pair-specific MCDI. The distance between two points corresponds to the fraction of neurons that responds differentially to each type of multipoint correlation. A semitransparent gray plane marks the 0-value along the vertical. Note that in V2, especially in the supragranular layer, there is a wide separation between *even* and *odd* stimuli, and between *black* and *white triangle* stimuli, and these separations lie on different axes.

The following figure supplement is available for figure 2:

**Figure supplement 1**. Sensitivity to multipoint correlations in V1 and V2 as a function of RF area and number of checks within the RF.

Sensitivity to multipoint correlations was not restricted to specific cell types. Specifically, the MCDI was not significantly associated with the simple vs complex distinction, nor with the distinction between regular-spiking and fast-spiking neurons, as determined by extracellular action potential shapes (*Niell and Stryker, 2008*). Sensitivity to multipoint correlations was also present in isolated units that did not have overt RFs by hand-mapping or by reverse correlation (81/269 in V1 and 65/152 in V2); these units had waveforms that were isolated by the tetrode recordings, and likely include many of the 'unresponsive' neurons (*Olshausen and Field, 2004*) that would not have been selected for study with single-electrode methods. There was no significant difference in the MCDI between these neurons and the simultaneously-recorded neurons with mappable RFs, either in V1 or V2. Among the neurons with mappable RFs, the MCDI was not significantly different for neurons whose RFs were above vs below the median size for their laminar compartment. Thus, the sensitivity to multipoint correlations does not require a precise match between the RF size and the spatial scale of the correlations. In sum, sensitivity to multipoint correlations was widely distributed across V2 neurons.

The difference in sensitivity to multipoint correlations between V1 and V2 was not due to a difference in RF size, nor to stimulus scaling (i.e., the number of stimulus checks within the RF). The upper left panel of *Figure 2—figure supplement 1* compares MCDI across V1 and V2 as a function of RF area; across the entire range of sizes, the MCDI is higher in V2 than in V1. The upper right panel makes this comparison as a function of the number of checks within the RF, which also equates neurons whose RFs covered the same fraction of the stimulus area. Here too, the MCDI in V2 was larger than in V1. The remaining rows of *Figure 2—figure supplement 1* break the analysis down by laminar compartment. In granular and supragranular layers, the above observations hold, but there is a suggestion of a subset of V1 neurons with large RFs (lower left panel) that are sensitive to multipoint correlations. However, it is unlikely that this subpopulation underlies the high MCDI seen in V2: the targets of infragranular V1 (*Felleman and Van Essen, 1991*) are the superior colliculus (layer 5) and the lateral geniculate (layer 6), while the inputs to V2 arise mainly from layers 2, 3, and 4b, where the MCDI is low. Moreover, the difference between supragranular V2 and the V2 input layer strongly suggests that the behavior in supragranular V2 is a result of intrinsic processing in V2, not a feature of signals passed on by V1 (which would already have been present in the granular layer).

*Figure 2B* shows that the multipoint correlations that contribute to the MCDI are the ones previously identified as being informative about natural images (*Tkačik et al., 2010*) and perceptually salient (*Victor and Conte, 1991*), namely, the *even, odd, white triangle*, and *black triangle* stimuli. *Figure 2C* further breaks down the MCDI into the individual pairwise comparisons. Few neurons, either in supragranular V2 or across the population, discriminated among pairs of the stimuli with uninformative multipoint correlations (*random, wye,* and *foot*). To visualize the pattern of discrimination across the neuronal population, we applied multidimensional scaling to the data of *Figure 2C*. This led to a three-dimensional representation (*Figure 2D*) in which the seven stimulus types are represented by points, and the distance between the points corresponds to the fraction of

neurons that distinguishes between them (i.e., the average pair-specific MCDI across the population). In V1, points are clustered near the origin, since most neurons cannot distinguish between any stimulus types. In V2, the representation expands into a multidimensional space. The two opposite stimulus pairs (*even* vs *odd*, and *white triangle* vs *black triangle*) are separated along different axes. Correspondingly, psychophysical studies show that the *even*-vs-*odd* gamut, and the *white triangle*-vs-*black triangle* gamut are independent perceptual axes (*Victor and Conte, 2012*) (Figure 8 panel 2 of reference [*Victor and Conte, 2012*], and the [$\theta$, $\alpha$]-panels of Figure 3 of *Hermundstad et al. (2014)*). Human perceptual sensitivities are larger for the four-point configuration than for the three-point configurations (*Victor and Conte, 2012*; *Hermundstad et al., 2014*); this is mirrored by higher values of the MCDI for the *even* stimuli than for the *white triangle* or *black triangle* stimuli in supragranular V2 (*Figure 2D*).

However, there are some differences between representation of informative multipoint correlations in the V2 population (as shown in *Figure 2D*) and human psychophysics (*Victor and Conte, 2012*; *Hermundstad et al., 2014*). First, the points corresponding to the uninformative stimuli are close to, but not superimposed on, the random stimulus. Additionally, while psychophysical sensitivity to the *odd* stimulus is only about 25% less than sensitivity to the *even* stimuli (*Victor and Conte, 2012*), the MCDI for the *odd* stimulus is much lower. We note that the *odd* stimulus contains *even* correlations when analyzed at spatial scales larger than a single check (*Victor and Conte, 1989*), so neuronal mechanisms sensitive to the *even* correlation will also contribute to the perceptual salience of the *odd* stimulus. More generally, the discrepancies between V2 neuronal activity and perception may reflect the simple measure used for quantifying discrimination at the population level (the average MCDI and multidimensional scaling), as well as further neural processing between V2 and perception.

## Discussion

Building on recent findings that the perceptual salience of complex (multipoint) image statistics is governed by their informativeness in natural images, here we show that selective sensitivity to these image statistics arises primarily in V2. Within V2, the greatest sensitivity is in the supragranular layers, where the typical (median) neuron can distinguish between two or three of the stimulus pairs. In contrast, typical neurons in V1 do not distinguish between any of the stimuli, although there appears to be a subpopulation of large-RF neurons in infragranular V1 with a modest level of selective sensitivity. The overall pattern of neuronal sensitivity to image statistics (*Figure 2D*) resembles the sensitivity of human observers, driven primarily by the multipoint statistics that are visually salient.

We speculate that sensitivity to informative multipoint correlations is the computational underpinning of many of the changes in neural characteristics from V1 to V2 that have previously been noted—sensitivity to corners, junctions (*Das and Gilbert, 1999*), illusory contours (*von der Heydt et al., 1984*), figure vs ground (*Qiu and von der Heydt, 2005*), and 'naturalness' (*Freeman et al., 2013*). The distinction between informative and uninformative multipoint correlations emerged from a formal information-theoretic analysis of natural images (*Tkačik et al., 2010*). While this analysis did not relate 'informativeness' to these other characteristics, inspection of the examples of *Figure 1* suggests several points of contact. With regard to junctions and contours, examples of the *odd* ensemble images contain large numbers of corners, while examples of the *even* ensemble contain large numbers of crossings. The extended contours evident in the *even* ensemble are a kind of illusory contour, since the polarity changes that define them undergo random flips, which would confound a linear edge detector. With regard to figure vs ground, stimuli in the *black triangle* and *white triangle* ensembles appear to contain, respectively, black figures on white backgrounds, vs white figures on black backgrounds—even though the stimulus sets are matched for spatial frequency content and the number of black and white checks. Thus, informative multipoint correlations result in images that are enriched for junctions, contours, and objects, compared to images that have the same first- and second-order statistics but lacking these correlations. While the extent to which these local features account for 'naturalness' remains for future work, the present results show that selective sensitivity of V2 neurons for informative multipoint correlations persists even when they are removed from the context of a natural image.

It is unclear to what extent it is necessary to match the scale of a multipoint correlation with that an illusory contour or junction in order for the visual feature to be extracted. However, the distinction between informative and uninformative multipoint statistics holds over at least a fourfold range of

length scales (the entire range analyzed, SI figure 14 of [*Tkačik et al., 2010*]). Human sensitivity to these correlations is present over at least a similar range of check sizes (0.03–0.25 deg, Figure 2, 8 of [*Victor and Conte, 1989*]; also [*Conte et al., 2014*]) comparable to the range of check sizes used in this study (0.08–0.5 deg). This broad range of sensitivities is found even when stimuli are restricted in eccentricity (*Victor and Conte, 1989*). *Figure 2—figure supplement 1* (right column) shows that V2 sensitivity to multipoint correlations also does not require a close match between RF size and the scale of the multipoint correlation; this sensitivity is present over a threefold range of length scales (i.e., a 10-fold range of the number of checks per receptive field). Thus, it is likely that the entire range of scales relevant to perception can be accounted for by the properties of individual neurons, along with the variation in RF sizes at each eccentricity (*Hubel and Wiesel, 1968*).

Neurons whose RFs are difficult to map are often ignored in physiologic studies (*Olshausen and Field, 2004*). We were able to analyze their responses here because of the tetrode recording method, and found that many V2 neurons whose RFs could not be mapped nevertheless often showed selective sensitivity to multipoint correlations. We consider some possible reasons for this here. As defined in this paper, a neuron is considered to have a mappable RF if the reverse correlation of the neuron's responses to the stimulus passes a statistical criterion (see 'Materials and methods'). Standard practice is to use random binary stimuli for this mapping procedure (*Reid et al., 1997*); here we include stimuli with high-order correlations in the mapping computation. The rationale is that inclusion of these stimuli allows some kinds of nonlinear responses to emerge in a first-order cross-correlation between stimulus and response, because of correlations within the stimuli (*Schmid et al., 2011*). But even an expanded stimulus set may not reveal the RFs of all neurons that respond to multipoint correlations. Reverse correlations may not exceed our statistical threshold because of response variability, or because the neuron is only responsive to stimulus configurations that occur very rarely in the stimulus set. We also note that from a computational point of view, our assay for sensitivity to multipoint correlations is independent of whether the neural response is correlated with the state of any single check: for each of the ensembles that probe a different kind of multipoint correlation, the number of black and white checks are equated, at each location. Thus, it is quite possible for a neuron to process information in a localized region of space (as manifest by its sensitivity to multipoint correlations) yet fail to have a RF that is measurable by reverse correlation methods, as we show here.

Finally, our findings carry implications for neural mechanisms. Many biologically-plausible mechanisms can extract multipoint correlations, including a simple linear-nonlinear cascade (provided that the nonlinearity is more than quadratic), and modulatory surrounds (*Schmid and Victor, 2014*; *Self et al., 2014*). But models need to account for the specificity of the responses, not just their existence. In this regard, we note (*Victor and Conte, 1991*) that the specificity we observe can be produced by a two-stage (linear-nonlinear-linear-nonlinear) cascade, in which the first linear-nonlinear element detects local edges, and the second one combines signals from collinear edges via a second threshold. Removal of either component of the second stage—either its linear or the nonlinear element—eliminates this specificity. The finding that responses to multipoint correlations are more prominent in supragranular V2 than in its input layers or in V1 suggests possible correspondences between this cascade and neural circuitry. One possibility is that the first stage is in V1 and the second stage is in V2 (*Wilson et al., 1992*; *Rust et al., 2005*). Alternatively, the two linear-nonlinear stages may represent two loops of signal passage through a recurrent network within a single cortical area (*Joukes et al., 2014*).

## Materials and methods

All procedures conformed to the guidelines provided by the US National Institutes of Health and Weill Cornell Medical College Animal Care and Use Committee. Full details concerning the physiologic preparation are provided in *Schmid et al. (2014)*, and are summarized here.

### Preparation

Single-unit recordings using arrays of three to six independently positioned tetrodes (typical input resistance, 1–2 MΩ; Thomas Recording GmbH, Giessen, Germany) were made in V1 and V2 of 14 macaques, anesthetized with propofol and sufentanil and paralyzed with vecuronium or rocuronium. Tetrodes were placed on opposite sides of the V1/V2 boundary, and typically within 1 mm of each other within each region, so that the units recorded by the tetrodes generally had neighboring or

overlapping RFs. This yielded a total dataset of 421 neurons (269 in V1, 152 in V2), following spike sorting and selection for firing rate criteria (see below).

## Initial neuronal characterization

Tetrodes were independently lowered until they recorded visually-driven extracellular action potentials. After initial hand-mapping, tuning properties were determined from responses to 3–4 s presentations of drifting sinusoidal gratings. Stimulus parameters were successively refined in the order of orientation, spatial frequency, temporal frequency and contrast, based on on-line analysis of the responses of the target unit. When the recorded cluster had well-isolated units that preferred an orientation other than the preferred orientation for the target unit, this process was repeated for a second, and rarely a third, orientation as well.

## Characterization of sensitivity to multipoint correlations

To determine neuronal responses to multipoint correlations, we measured responses to a sequence of black-and-white checkerboards that isolated the individual kinds of correlation. *Figure 1* (*top*) shows three examples of these seven stimulus types. Each stimulus consisted of a 16 × 16 array of black and white checks. In the '*random*' stimulus set, check colors were assigned independently, with an equal chance of being white or black. In the other stimulus sets, the coloring rule isolated a single kind of multipoint correlation. In the '*even*' set, there was always an even number of white (or black) checks in any 2 × 2 neighborhood of checks. In the '*odd*' set, there was always an odd number of white (or black) checks in a 2 × 2 neighborhood. *Even* and *odd* sets are the opposite extremes of the visually salient four-point correlation (*Hermundstad et al., 2014*), α. In the '*white triangle*' set, there were always one or three white checks within a triangular region; in the '*black triangle*' set, there were always one or three black checks within a region of the same shape. These two sets correspond to opposite extremes of the visually salient three-point correlation (*Hermundstad et al., 2014*), θ. We also examined responses to four-point correlations in two other spatial configurations, '*wye*' and '*foot*.' Multipoint correlations in the *wye* and *foot* configurations are predictable from simpler quantities in natural images (*Tkačik et al., 2010*), and, in keeping with predictions of efficient coding (*Hermundstad et al., 2014*), they are not visually salient (*Victor and Conte, 1991*).

Check size was scaled to the RF size of the target neuron so that approximately two checks corresponded to one lobe of the optimal spatial frequency, and orientation was set according to the orientation preference of the target neuron. This resulted in about 8 checks within the classical RF (V1: mean 7.40, median 6.00, SD 5.33; V2: mean 8.68, median 7.00, SD 5.46; statistics across all mappable units and not just the target; see below for details on RF mapping and *Figure 2—figure supplement 1* for the distribution of number of checks in the RF); thus, the checks are within the resolution limits of the neuron, and the stimulus patch covers an area that is substantially larger than the RF. Across all recordings (including mappable and un-mappable units), check sizes ranged from 0.08 to 0.5 deg (V1: mean 0.18, median 0.20, SD 0.05; V2: mean 0.22, median 0.20, SD 0.12).

For each type of stimulus, we presented 1024 examples (two repetitions each) for 320 ms, interleaved in a pseudorandom sequence. This large set size was chosen so that we can distinguish average responses to each of the stimulus sets (our focus) from responses that might be driven to the specific white or black checks or edges present in particular examples (a potential confound).

Stimuli were generated via a Markov recurrence rule (*Victor and Conte, 1991*, *2012*), so that other than the constraint of their defining multipoint correlation, they are as random as possible (maximum-entropy). This yields stimulus sets that enable testing of each kind of multipoint correlation in isolation. In each set, there are no two-point correlations—checks at any pair of locations are colored independently—so that the sets have the same power spectra, and therefore the same spatial frequency content. The four kinds of correlations (the *even/odd* axis, the *black triangle/white triangle* axis, *wye*, and *foot*) are independently controlled: each set extremizes one of these correlations, while keeping all the others at 0 (*Gilbert, 1980*). Thus they provide a way to assay responsiveness to each kind of multipoint correlation in isolation.

All stimuli were rendered on a 1280 × 1024-pixel display at 100 Hz, using either a 21-inch ViewSonic G225f monitor (mean luminance 47 cd/m², gamma-corrected) or a Sun GDM5410 monitor (mean luminance 46 cd/m², gamma-corrected) at 114 cm. Control signals for the displays were generated by PC-based system using OpenGL software.

## Spike sorting

After bandpass filtering (300–9000 Hz) and thresholding, waveforms were clustered using customized versions of KlustaKwik and Klusters (*Hazan et al., 2006*); details as in *Schmid et al. (2014)*. The 17 features consisted of peak and trough amplitudes (8 features), the first 8 principal components, and time. All neurons whose mean firing rates across all stimuli were ≥ 1 Hz were analyzed for their responsiveness to the multipoint correlation stimuli described above.

To classify extracellular spike waveforms as narrow-spiking (putative inhibitory) and broad-spiking (putative excitatory), we used a method similar to that of refs. (*Mitchell et al., 2007*) and (*Niell and Stryker, 2008*). For each single unit, the waveforms from each tetrode channel were averaged and the channel with the largest signal to noise ratio (SNR) was selected for the spike width measurement. Two parameters of spike width were measured: (1) trough to peak width—the duration from the trough to the peak of the waveform, and (2) half-peak width—the duration from the peak of the waveform to half its height. The distribution of both measurements across the 1856 waveforms from the laboratory database were significantly bimodal ($p < 0.01$ by the Hartigan dip test [*Hartigan and Hartigan, 1985*]). Based on the notch in the distribution, we classified extracellular waveforms as narrow-spiking (<405 µs) and broad-spiking (>430 µs). Next the averaged waveforms themselves were clustered using k-means. The clusters were separated identically by k-means of the waveforms, and the distribution of the spike width parameters.

## Localization of recording sites

At the conclusion of the experiment, we made small lesions at locations that bracketed the recording sites along each tetrode track, via current passage through the most distal tetrode contact. Details concerning the procedures for lesions, perfusion, and histology are in ref. (*Schmid et al., 2014*). For sites for which the laminar assignment was uncertain, neurons were included in the tallies for V1 and V2 (e.g., top rows of each panel of *Figure 2*) and the statistical comparisons between them, but not in the breakdown by layer or statistical comparisons between layers. This amounted to <10% of the units.

## Data analysis

Tuning curves were computed in the standard fashion from the Fourier components of the spike train elicited by each grating stimulus, as detailed in *Schmid et al. (2014)*. Tuning curve peaks were determined from the DC response (F0) or the first harmonic (F1), whichever was larger. We classified neurons as simple or complex according to whether their response to a drifting grating was primarily at the period of the grating (simple) or primarily a maintained elevation (complex), as quantified by the F1/F0 ratio (*Skottun et al., 1991*). F1 is the first harmonic of the response to the optimal grating tested, F0 is the maintained firing rate of the response, after subtraction of the average firing rate in response to a uniform field at the mean illumination. Note that since grating parameters were chosen according to the preference of neurons whose waveforms could be discriminated online, some neurons may not have been stimulated at the optimal orientation or spatial frequency.

RF maps were determined by correlating the neural response (1 for white checks, −1 for black checks) to the checkerboard stimuli (16 × 16 checks). The response measure was the total number of spikes over the duration of each presentation (320 ms) averaged across both repetitions; this is equivalent to computing the spike-triggered-average and then summing over the stimulus duration. Maps were separately computed for each of the seven stimulus types; as reported previously, some neurons (*Schmid et al., 2012*) that did not have mappable RFs for random checkerboards nevertheless had mappable RFs for the other stimulus types. Statistical significance for each of these seven maps was determined by a shuffle test: we recomputed maps from 500 surrogate data sets in which the responses to each stimulus type were permuted, determined the mean and standard deviation of these surrogate maps at each check, and then used the corresponding Gaussian distribution to determine which actual map values were significant at $p < 0.05$ (two-tailed, correcting for multiple comparison via the Benjamini-Hochberg method, that is, false discovery rate [FDR] method) (*Benjamini and Hochberg, 1995*). We then determined the union of the seven maps obtained from each stimulus. Usually this yielded a single connected component, and the RF was taken to be its convex hull. When more than one connected component was present, smaller components were merged with the largest one if they were separated by no more than a single check,

and the convex hull of the resulting region was taken as the RF. The number of checks in this convex hull was taken as the measure of RF size. If none of the seven classes of stimuli yielded a significant RF map by the above criteria, the neuron was considered not to have a mappable RF. As an alternative procedure, we also computed RF maps by correlating the responses with all (7 × 1024) stimuli, and this yielded very similar results.

To measure sensitivity to multipoint correlations, we proceeded as follows (see *Figure 3*). For each of the stimulus types, we accumulated a PSTH across all 1024 examples (and 2 repeats), and then determined the smoothed firing rate via local linear regression (*Loader, 2012*). Significance of the difference between two firing rate functions was determined by a shuffle test, in which 3000 surrogate data sets were created by randomly exchanging responses among a pair of stimulus types. The exchanges were limited to responses that were recorded in adjacent trials (within 4 s of each other), to avoid confounds due to slow changes in firing rate over time. The difference

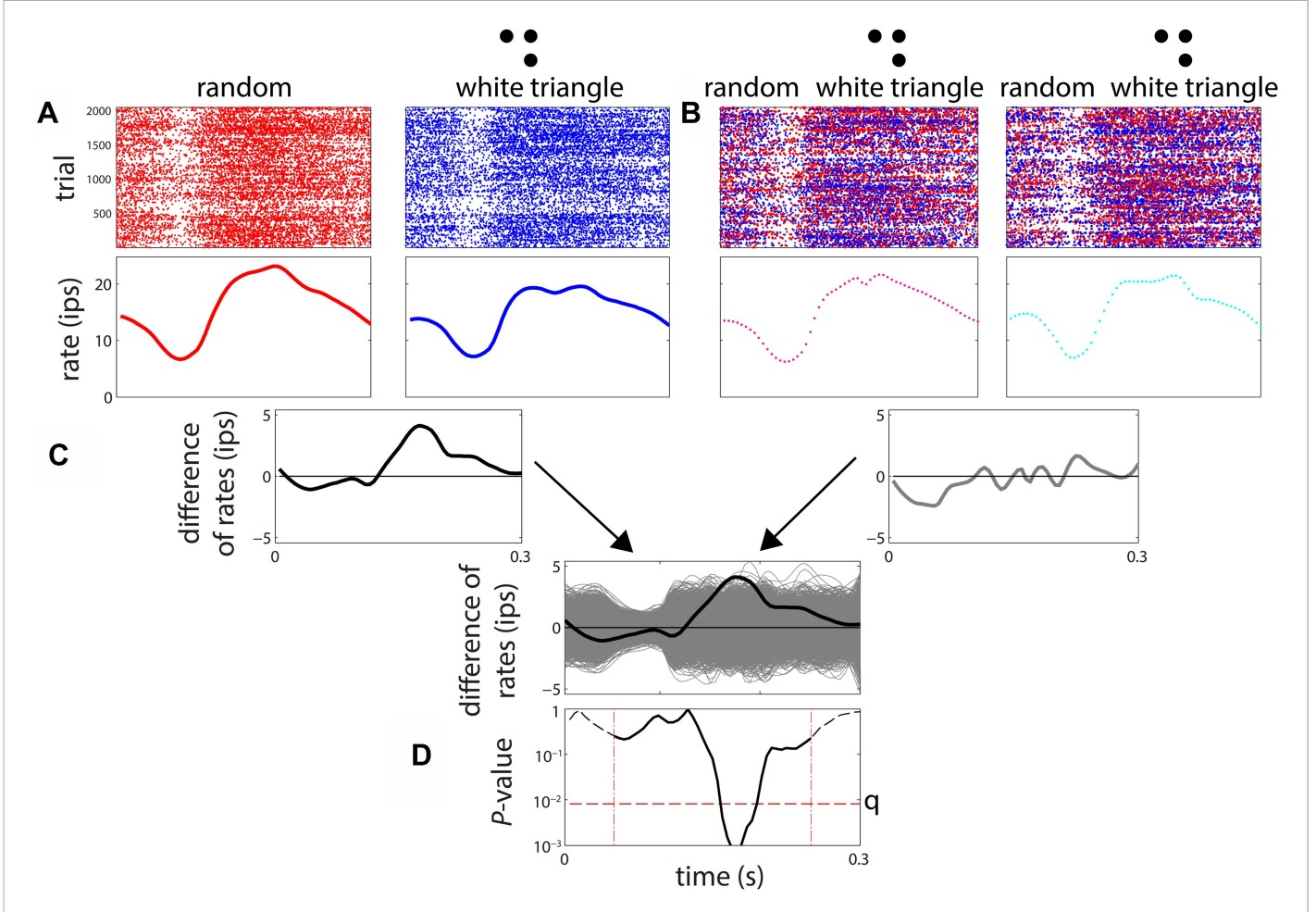

**Figure 3**. Procedure for determination of differential sensitivity to multipoint correlations for a stimulus pair. (**A**) A smoothed firing rate is constructed from the responses to examples of each stimulus type (1024 examples, each presented twice). (**B**) A parallel procedure is carried out for 3000 surrogate datasets, in which responses are randomly exchanged among the stimuli. The exchanges were limited to responses recorded in adjacent trials, to avoid confounds due to slow change in firing rate over time. (**C**) The difference between the smoothed responses to the two stimuli is computed, both for the actual responses and each of the surrogate datasets. The relationship of the actual firing rate difference (black) to the distribution of differences encountered in the surrogate datasets (gray) is determined. (**D**) At each time point, the position of the actual difference in the surrogate difference distribution is expressed as a two-tailed p-value. The actual difference is considered to be significant if any of these p-values over the range 55–250 ms (dashed vertical lines) fall below the false-discovery-rate (FDR) threshold q corresponding to a significance level of 0.05. The FDR threshold q, illustrated as the horizontal dashed line in *Figure 3D*, is a data-determined quantity (*Benjamini and Hochberg, 1995*) that is substantially less than the raw significance level of 0.05 (in this case, q < 0.01).

between the smooth firing rates of the actual data was compared to the distribution of differences seen in the surrogate datasets at each 5 ms bin, from 55 to 250 ms. The number of times the actual difference was exceeded by any of the 3000 surrogates yielded a raw two-tailed p-value at each of these 40 time points. If the raw p-value was below the false-discovery-corrected threshold of p = 0.05, the neuron was considered to have a different response to the two kinds of stimuli at that time point. For each neuron, the MCDI at each time point (*Figure 2A*) was defined as the fraction of stimulus pairs that elicited statistically different responses as determined by the above procedure; the MCDI was therefore $n/21$, where $n$ is the number of stimulus pairs that elicited statistically different responses. For each of the seven stimuli, we also calculated a stimulus-specific MCDI, considering only the six pairs of discriminations involving that particular stimulus (*Figure 2B*); this was therefore a quantity $n/6$. Finally, to detail the pattern of pairwise discriminations (*Figure 2C,D*), we computed a 'pair-specific' MCDI—either 0 (no discrimination) or 1 (discrimination), and averaged it across the population. For this purpose, we considered a neuron to distinguish a pair of stimuli if a difference was present at any time during the 55–250 ms period, again using the above statistical criteria.

Sensitivity to multipoint correlations was not associated with the simple vs complex distinction (as measured by F1/F0 ratio, with a dividing point at 1 [as in ref. *Mechler and Ringach, 2002*] or at the population median). These and other comparisons between subsets of cells (e.g., V1 vs V2, or between laminar compartments) were carried out using a two-tailed Wilcoxon rank-sum test. The raw p-values were subjected to false discovery correction (*Benjamini and Hochberg, 1995*) across time points, in 5 ms bins from 55 ms to 250 ms. Statistical significance corresponds to $p < 0.05$.

To visualize the population pattern of differential responses (*Figure 2D*), we used standard multidimensional scaling (*Kruskal and Wish, 1978*), applied to the fraction of neurons that distinguished between each pair of correlation types. The first two embedding dimensions (as shown in *Figure 2D*) typically accounted for > 90% of the variance.

## Acknowledgements

This work was supported by NIH EY09314 to JV. We thank Ferenc Mechler, Ifije Ohiorhenuan, Qin Hu and Eyal Nitzany for their assistance with the physiological experiments, Mary Conte and Keith Purpura for many helpful discussions, and Ann Hermundstad for her comments on the manuscript. We also thank Daniel Thengone for his help classifying spike waveforms into narrow- and broad-spiking. A portion of this work was reported at the 2013 meeting of CoSyNe (Computational and Systems Neuroscience), Salt Lake City, UT.

## Additional information

### Funding

| Funder | Grant reference | Author |
|---|---|---|
| National Institutes of Health (NIH) | EY09314 | Jonathan D Victor |

The funder had no role in study design, data collection and interpretation, or the decision to submit the work for publication.

### Author contributions

YY, JDV, Conception and design, Acquisition of data, Analysis and interpretation of data, Drafting or revising the article; AMS, Acquisition of data, Drafting or revising the article

### Author ORCIDs

Jonathan D Victor, http://orcid.org/0000-0002-9293-0111

### Ethics

Animal experimentation: This study was performed in strict accordance with the recommendations in the Guide for the Care and Use of Laboratory Animals of the National Institutes of Health. All animal procedures were approved by the Institutional Animal Care and Use Committee (IACUC) of Weill Cornell Medical College, protocol 0810-795A.

# Additional files

## Supplementary file
• Source code 1. Custom software written in MATLAB. All computations were done with custom software written in MATLAB (The MathWorks, Inc. MA). In *Source code 1*, we provide the source code we used for smoothed firing rate computations, computation of the MCDI, statistics, and plotting. Note, though, that much of this code is tied to the specifics of our file formats, naming conventions, etc., and that locfit, used for smoothed firing rate computations, should be downloaded from http://cran.r-project.org/web/packages/locfit/index.html, and compiled for the target system. As mentioned above, spike-sorting was done with Klustakwik and Klusters; these packages can be downloaded from http://klustakwik.sourceforge.net/ (Klustakwik) and http://neurosuite.sourceforge.net/ (Klusters), and then compiled for the target system. *Source code 1* is also held on figshare under doi: 10.6084/m9.figshare.1409463.

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
