## [Decision Letter]

Thank you for sending your work entitled “Visual processing of informative multipoint correlations arises primarily in V2” for consideration at *eLife*. Your article has been favorably evaluated by Timothy Behrens (Senior editor and Reviewing editor) and three reviewers, one of whom, Michael Landy, has agreed to share his identity.

The editor and the reviewers discussed their comments before we reached this decision, and the editor has assembled the following comments to help you prepare a revised submission.

The editor and reviewers agree that the finding of neural correlates of multipoint correlations reflects an important advance over your previous behavioural findings and are enthusiastic about the potential publication of this Research Advance.

For example:

This paper takes the previous work of this group on which 2-, 3- and 4-point correlations are visually salient and to which the visual system is sensitive, and shows that some differential responses to these correlations arise first in area V2. As the authors are aware, I'm pretty familiar with this line of work (having reviewed several of the earlier papers in the series). Tying this story to the physiology is certainly a logical and useful next step.

In the current manuscript Yu et al. add to previously published findings that human observes are more sensitive to more informative multi-point correlations in images. Here they provide a candidate for the neural substrate of this sensitivity: supragranular layers of V2. I think the manuscript is an interesting addition to the previous paper.

Overall, this Research Advance is clearly written and nicely complements the founding article by providing neuronal correlates for multipoint correlation stimuli that have theoretical significance and perceptual relevance.

However, there were several questions that the review panel would like addressed before we could consider publication of the study.

During discussion the panel agreed that the most critical issue to address before the paper can be published is the issue of the scaling of the stimuli.

Since receptive fields are larger in V2 and you are adjusting the stimulus to the receptive field size, aren't you effectively presenting two different stimulus ensembles to the V1 and the V2 population, respectively? Could that explain the differential response between V1 and V2? Do you have control data, where you, instead of upscaling the 16x16 patch, simply increased the number of pixels to match the receptive field size? Alternatively, you could drive the V1 population with the upscaled stimuli for the V2 neurons and see whether your results change. If you do not have such data readily to hand, do you have other means of ruling out that the stimulus scaling confounds the results? For example, are there recorded V1 cells whose receptive field completely overlapped with that of a V2 cell? In that case, you would have responses to the same stimulus ensemble, but from two different areas. The review panel agreed that this issue should be rigorously addressed.

If such data do not exist, then the review panel asks you to remove the claim of a distinction between V1 and V2 from the paper since the data does not really support a comparison, and to explicitly mention that V1 and V2 are stimulated with differently scaled stimuli and explain the reasoning behind it.

A related question about the relationship between V1 and V2 coding was also raised:

Figure 2 shows that at least 75% of V1 cells have no selectivity for multipoint correlation stimuli, yet the mean MCDI of all cells is ∼0.05. This implies that V1 has a small population of V1 cells that have an MCDI of 0.2 or more. How do the response properties of these “special” V1 neurons compare to a “typical” V2 neuron? With the current presentation, it's hard to tell whether V2's representation of multipoint correlations is new or enhances a representation already present in a small subpopulation of V1. As such, the first sentence of the Discussion, which says “arises primarily in V2” seems imprecise. This distinction is also relevant to the authors' Discussion hypothesis that higher-order correlation specificity might emerge from a two-stage cascade from V1 to V2.

The reviewers were also concerned about the illusory contour figure:

At present, the connections between higher-order correlations and previously hypothesized roles for V2 seem tenuous and insufficiently detailed to warrant a full figure in the main text.

For example:

The authors use Figure 3 to argue that V2's selectivity to multipoint correlations helps explain its involvement in the detection of illusory contours and the discrimination of figure and ground. I have two comments. First, in panel a, the “even and odd” correlation structure picks out the corners of the black bars. As such, the association of this correlation with the illusory contour is a consequence of the fact that the bars are spaced by the same distance that defined the “even and odd” correlation structure. Can the authors say anything about whether there is an association between the spatial scales of illusory contour detection and the “even and odd” correlation structure? For example, do humans perceive illusory contours over the same length scales that “even and odd” correlation structures are informative for natural images? Has V2 previously been shown to be sensitive to illusory contours on the spatial scale that the authors use in the “even and odd” correlation structure? Second, the “white and black triangle” correlation structure is the only correlation structure that can distinguish between the two stimuli in panel b because it's the only odd-ordered correlation. This is why I earlier alluded to the point that it would have been helpful to include an uninformative third-order correlation stimulus. Also, in the specific example shown, couldn't one just use the mean (i.e. a first-order structure) to discriminate between the stimuli? I wonder if there might be a better choice of stimuli for this panel.

The reviewers also had several questions that we believe can be addressed by changes to the manuscript text.

In [11] the second order stimuli (beta) are more informative than the fourth order stimuli (alpha), which are more informative than the third order stimuli (theta). However, the authors only present data for alpha and theta stimuli (that were less informative in the Hermundstad paper) while not presenting the beta stimuli (that were more informative in the Hermundstad paper). I would like to know whether the authors (i) performed experiments with beta stimuli, (ii) if so why they did not report them, or (iii) why they did not consider them. I am sure the authors had a good reason which should be mentioned in the paper.

It would be nice if you could mention more explicitly how the rank order of the MCDI relate to the sensitivity order found in the psychophysical experiments of the previous paper.

I was interested by the finding that the MCDI was unassociated with mappable receptive fields and would be interested in hearing some thoughts from the authors in their Discussion section.

Can the authors clarify how they determined the p-value threshold in Figure 4D? Since the authors declare significance “if any of these p-values” falls below the FDR threshold, is the Benjamini-Hochberg correction equivalent to a Bonferroni correction? If so, then 40 comparisons would lead to a p-value correction less than that displayed. Or do the authors somehow correct for fact that their temporal smoothing effectively leads to fewer than 40 comparisons?

There were also questions about the underlying assumptions in the model. We understand that these comments pertain equally to the already-published paper, but we nevertheless hope that you will be able to deal with them in a few sentences, which we felt would help the current manuscript.

What justifies the authors to assume the regime of sampling limitation rather than transmission limitation? You write that your results fit into the efficient coding framework if sampling an image is the main limitation. What is the empirical evidence that justifies this assumption as opposed to the transmission limited regime many other studies are based on?

If I understand correctly, the fact that humans/neurons should be more sensitive to more variable features is derived using a linear model with Gaussian input and channel noise. However, the mapping from images to multi-point correlations does not seem to be linear. How do you know that this result still holds in the non-linear case, in particular if the sampling regime is characterized by dominating input noise (which would get nonlinearly transformed)?

How do you justify that more variable features contain more information? In the discrete case, I can see that. However, in the limit of infinitely many images, the multi-point correlations become continuous. In that case I could transform all features by a pointwise monotonic transformation (histogram equalization) that would not change the information content but make all features equally variable.

---

## [Author Response]

*During discussion the panel agreed that the most critical issue to address before the paper can be published is the issue of the scaling of the stimuli*.

*Since receptive fields are larger in V2 and you are adjusting the stimulus to the receptive field size, aren't you effectively presenting two different stimulus ensembles to the V1 and the V2 population, respectively? Could that explain the differential response between V1 and V2? Do you have control data, where you, instead of upscaling the 16x16 patch, simply increased the number of pixels to match the receptive field size? Alternatively, you could drive the V1 population with the upscaled stimuli for the V2 neurons and see whether your results change. If you do not have such data readily to hand, do you have other means of ruling out that the stimulus scaling confounds the results? For example, are there recorded V1 cells whose receptive field completely overlapped with that of a V2 cell? In that case, you would have responses to the same stimulus ensemble, but from two different areas. The review panel agreed that this issue should be rigorously addressed*.

*If such data do not exist, then the review panel asks you to remove the claim of a distinction between V1 and V2 from the paper since the data does not really support a comparison, and to explicitly mention that V1 and V2 are stimulated with differently scaled stimuli and explain the reasoning behind it*.

We agree that this is an important issue, and we also very much appreciate the latitude given by the reviewers in responding to the point. Because of space limitations in the original submission, we had only touched on this, mentioning that responsiveness to multipoint correlations was similar for neurons with large and small receptive fields (RF’s); we now devote a new figure to this issue (Figure 2—figure supplement 1), and add material describing this analysis to the Results section, after Figure 2. Briefly, the new figure shows that the difference between V1 and V2 cannot be explained merely by a difference in RF size, or a difference in the number of checks within the RF: whether one compares the MCDI for V1 and V2 neurons with comparable RF sizes, or for V1 and V2 neurons stimulated with comparably scaled stimuli, the V2 neurons have a larger MCDI. The upper left panel shows the MCDI is plotted as a function of RF area, showing a difference in MCDI for V2 vs. V1 neurons across the entire range. So, V2 neurons are not simply larger versions of V1 neurons. But, as the reviewers point out, since we adjust the stimulus to match the resolution of the neural activity at one of the tetrodes (i.e., at one brain site), there is the possibility that the V2 vs. V1 difference relates to the number of the checks in the receptive field, or, equivalently, the fraction of the 16 x 16 stimulus array that is “seen” by the RF. The upper right panel rules out this possibility as well: when compared on the basis of equal number of checks within the RF, V2 neurons have a larger MCDI than V1 neurons. The remaining rows of Figure 2 supplement break the analysis down by laminar compartment; there are no surprises in the supra‐ and granular layers but the analysis of the infragranular compartment (see below) suggests that there is a subset of V1 neurons with large receptive fields that are sensitive to multipoint correlations. We now mention this point at the end of the new paragraph devoted to Figure 2—figure supplement 1, but (as discussed below) it does not change our basic finding, that sensitivity to multipoint correlations arises primarily in V2.

We also want to mention (see point below concerning number of checks in the RF) that the number of checks considered to be within the RF (median, ∼8) is probably an underestimate for the actual RF size, as a check is only to be considered to be “within” the RF if reverse‐correlation analysis passes a criterion level of statistical significance; a substantial fraction of the dataset has neurons that did not meet this criterion but still had clear responses to the stimuli, as manifest by a large MCDI.

*A related question about the relationship between V1 and V2 coding was also raised*:

Figure 2
*shows that at least 75% of V1 cells have no selectivity for multipoint correlation stimuli, yet the mean MCDI of all cells is ∼0.05. This implies that V1 has a small population of V1 cells that have an MCDI of 0.2 or more. How do the response properties of these* “*special*” *V1 neurons compare to a* “*typical*” *V2 neuron? With the current presentation, it's hard to tell whether V2's representation of multipoint correlations is new or enhances a representation already present in a small subpopulation of V1. As such, the first sentence of the Discussion, which says* “*arises primarily in V2*” *seems imprecise. This distinction is also relevant to the authors' Discussion hypothesis that higher-order correlation specificity might emerge from a two-stage cascade from V1 to V2*.

There are two related points here, and both are interesting: (a), given that average MCDI in V1 is driven by a small subset of cells with large MCDI’s, does this subset constitute a subpopulation with definable features? And (b), should one view V2 as merely enhancing a representation of multipoint correlations that are already present in V1? With regard to (a), the new analysis presented in Figure 2—figure supplement 1 provides an important clue, which we now mention in the Results where the summary statistics are described: the infragranular layer contains a subset of V1 neurons, with large receptive fields, and this subpopulation has a large MCDI. With regard to (b), we don’t think that the existence of these neurons changes the basic conclusion that sensitivity to multipoint correlations arises primarily in V2: these units make up only a small subset of one laminar compartment in V1, and even among these neurons, the median MCDI does not approach the levels in supragranular V2. Independently, anatomical evidence supports this view: the targets of infragranular V1 (6) are the superior colliculus (layer 5) and the lateral geniculate (Layer 6); while the inputs to V2 arise mainly from layers 2, 3, and 4b, where the MCDI is low. Finally, the difference between supragranular V2 and the V2 input layer strongly suggests that the behavior in supragranular V2 is a result of intrinsic processing in V2, not a feature of signals passed on by V1 (which would already have been present in the granular layer). We now mention these points in the main text concerning Figure 2—figure supplement 1. Finally, to add precision to the broad statement that sensitivity to multipoint correlations arises primarily in V2, we add mention of the subpopulation of infragranular V1 neurons to the opening paragraph of the Discussion.

*The reviewers were also concerned about the illusory contour figure*:

*At present, the connections between higher-order correlations and previously hypothesized roles for V2 seem tenuous and insufficiently detailed to warrant a full figure in the main text*.

Our main reason for presenting this figure was to provide an intuition as to why only some kinds of multipoint correlations are informative, as an aid to readers who might not be comfortable with the technicalities of the approach taken in Hermundstad et al. and Tkacik et al. (PNAS, 2010). But we agree that the argument made by the figure is intuitive and un‐rigorous, rendering it unhelpful for the more sophisticated reader, so we removed it. Nevertheless it is useful to speculate about why these multipoint correlations are important, and we retain these comments, which we clearly mark as speculative. These comments, which are in a rewritten second paragraph of the Discussion, use the stimulus examples in the retained Figure 1 to illustrate the points made.

*For example*:

*The authors use*
Figure 3
*to argue that V2's selectivity to multipoint correlations helps explain its involvement in the detection of illusory contours and the discrimination of figure and ground. I have two comments. First, in panel a, the* “*even and odd*” *correlation structure picks out the corners of the black bars. As such, the association of this correlation with the illusory contour is a consequence of the fact that the bars are spaced by the same distance that defined the* “*even and odd*” *correlation structure. Can the authors say anything about whether there is an association between the spatial scales of illusory contour detection and the* “*even and odd*” *correlation structure? For example, do humans perceive illusory contours over the same length scales that* “*even and odd*” *correlation structures are informative for natural images? Has V2 previously been shown to be sensitive to illusory contours on the spatial scale that the authors use in the* “*even and odd*” *correlation structure*?

With regard to the scaling question: we don’t know to what extent it is necessary to match the scale of a multipoint correlation with the features that define an illusory contour in order for the contour to be extracted. As the reviewer notes, the former Figure 3 suggests that this is the case, but of course one expects that illusory contours are present across a wide range of scales too.

On the other hand, our findings do not rest on a narrow range of RF sizes or correlation scales, and the broad range of spatial scales to which neurons are sensitive to multipoint correlations corresponds to that of human perceptual sensitivity. Specifically, the distinction between informative and uninformative four‐point correlations persists across over a fourfold range of length scales (the entire range analyzed, see SI of Tkacik et al., PNAS 2010, Figure 14). The characteristics of human perception are approximately constant over a similar range of length scales (.03 to 0.25 deg checks, a range similar to that used in these experiments (Figure 2 and 8 of [32] [four‐point correlations]; also Victor, Thengone, Rizvi, and Conte et al., VSS Abstracts 2014 [two‐ and three‐point correlations as well]). Sensitivity across this range of scales is present even when eccentricity is restricted (32). While individual neurons may not subtend the entire range, the population of V1/V2 neurons as a whole is likely to do, and Figure 2—figure supplement 1 shows that V2 sensitivity to multipoint correlations does not require a close match between RF size and the scale of the correlations. We now add a paragraph about this to the Discussion, and add further specifics about check size to the Methods section.

*Second, the* “*white and black triangle*” *correlation structure is the only correlation structure that can distinguish between the two stimuli in panel b because it's the only odd-ordered correlation. This is why I earlier alluded to the point that it would have been helpful to include an uninformative third-order correlation stimulus. Also, in the specific example shown, couldn't one just use the mean (i.e. a first-order structure) to discriminate between the stimuli? I wonder if there might be a better choice of stimuli for this panel*.

The “earlier alluded” passage seems to have been omitted from the consolidated reviews, but we think we understand the point. We agree that it would have been a good control to include an uninformative third‐order correlation, but previous work on natural image statistics did not identify any such configurations (and didn’t seek to do so). We agree with the comment that other third‐order correlations could distinguish figure from ground. But the reviewer is incorrect about first‐order statistics—they would not have been able to make this distinction, as the fraction of black and white checks in the figure/ground component of the former Figure 3 was each 50% (this is not at all apparent to casual viewing).

*The reviewers also had several questions that we believe can be addressed by changes to the manuscript text*.

*In*
[11]
*the second order stimuli (beta) are more informative than the fourth order stimuli (alpha), which are more informative than the third order stimuli (theta). However, the authors only present data for alpha and theta stimuli (that were less informative in the Hermundstad paper) while not presenting the beta stimuli (that were more informative in the Hermundstad paper). I would like to know whether the authors (i) performed experiments with beta stimuli, (ii) if so why they did not report them, or (iii) why they did not consider them. I am sure the authors had a good reason which should be mentioned in the paper*.

The reason that we did not include them in the paper was that our focus was on determining the origin of sensitivity to three‐ and four‐point statistics, and the study was designed to concentrate the experiment time on that issue. Second‐ and first‐order statistics, which affect, respectively, the mean luminance and spatial frequency content of the stimuli, will modulate responses even at the level of retinal ganglion cells, so clearly there’s no mystery as to where responses to such statistics arise. We now state this rationale in the first paragraph of Results.

On the other hand, we agree that the extent to which the pattern of sensitivities in V1 and V2 accounts for the pattern of human sensitivities to local image statistics of all orders (e.g., the shape and orientation of the isodiscrimination contours in Hermundstad et al.) is interesting and important, and we are carrying out experiments in the macaque to look at this. Preliminary results confirm the expectation of sensitivity to first‐ and second‐order statistics in V1 and V2 (as well as the current findings about third‐ and fourth‐order sensitivities primarily in V2), and also indicate that neurons have preferences pointing in many oblique directions in the space. But this goes far beyond the current manuscript, and we think deserves its own paper.

*It would be nice if you could mention more explicitly how the rank order of the MCDI relate to the sensitivity order found in the psychophysical experiments of the previous paper*.

We now add this material to the final two paragraphs of the Results section, underscoring that the V2 results are not a precise match for perception, but this is not surprising, as further processing may well intervene.

*I was interested by the finding that the MCDI was unassociated with mappable receptive fields and would be interested in hearing some thoughts from the authors in their Discussion section*.

We add a paragraph concerning this to the Discussion. In brief, there are several ways in which neurons that are not “mappable” could, nevertheless, participate in form vision tasks in general (19) , or, in particular, discriminate among multipoint correlations (as we show here). As defined in this paper, a “mappable” RF requires that reverse correlation of the neuron’s responses to the stimulus passes a statistical criterion. Standard practice is to use random binary stimuli for this mapping procedure; here we include stimuli with high‐order correlations in the mapping computation. The rationale is that this allows some kinds of nonlinear responses to emerge in a first‐order cross‐ correlation between stimulus and response, because of correlations within the stimuli (Schmid, Yu, and Victor, SfN Abstracts, 2012). But there is no reason to believe that this expanded stimulus set reveals all RFs. For example: (i) reverse correlations may not exceed threshold because of response variability; (ii) a neuron may only be responsive to stimulus configurations that occur very rarely in the stimulus set; or (iii) a neuron may be selectively responsive to one or more of the multipoint correlations tested, but since the occurrence of this configuration is not correlated with whether a check is white or black, there will be no correlation of the response to the state of individual checks. Our data do not permit us to determine which of these explanations dominate, but we suspect that (i) and (iii) contribute.

*Can the authors clarify how they determined the p-value threshold in Figure 4D? Since the authors declare significance* “*if any of these p-values*” *falls below the FDR threshold, is the Benjamini-Hochberg correction equivalent to a Bonferroni correction? If so, then 40 comparisons would lead to a p-value correction less than that displayed. Or do the authors somehow correct for fact that their temporal smoothing effectively leads to fewer than 40 comparisons*?

Sorry for being unclear on this. As is standard, the FDR threshold for a given significance level alpha is a data‐ determined quantity q, for which it can be anticipated that less than a fraction alpha of the values below q are false‐ positives. Typically q is substantially less than alpha (in the case shown in the Figure, alpha is 0.05 but q is <0.01). We clarify this by labeling the significance threshold in panel d as “q”, and adding text to the caption to highlight that the FDR threshold q is distinct from, and more stringent than, the un‐corrected significance threshold of alpha=0.05. With regard to positive correlations among the individual values, the false discovery correction procedure we used takes this into account: under conditions of positive correlation, the standard procedure reduces to the FDR procedure we used (Benjamini & Yekutieli, (2001), “The control of the false discovery rate in multiple testing under dependency”, Annals of Statistics 29 (4): 1165–1188. doi:10.1214/aos/1013699998. MR 1869245). [Note: previous Figure 4, the subject of these changes, is now Figure 3.]

*There were also questions about the underlying assumptions in the model. We understand that these comments pertain equally to the already-published paper, but we nevertheless hope that you will be able to deal with them in a few sentences, which we felt would help the current manuscript*.

*What justifies the authors to assume the regime of sampling limitation rather than transmission limitation? You write that your results fit into the efficient coding framework if sampling an image is the main limitation. What is the empirical evidence that justifies this assumption as opposed to the transmission limited regime many other studies are based on*?

The empirical evidence for the “sampling limitation” hypothesis is that it predicts the main finding of Hermundstad et al., relating psychophysical sensitivities of local features to statistics of natural images: greater sensitivity for image statistics that are more variable. Had output capacity (transmission) been the main limitation, psychophysical sensitivities would have been large for image statistics that were less variable—the opposite of what we found (Figure 4B vs. Figure 4C of Hermundstad et al.). We now add this point to the first paragraph of the Introduction.

*If I understand correctly, the fact that humans/neurons should be more sensitive to more variable features is derived using a linear model with Gaussian input and channel noise. However, the mapping from images to multi-point correlations does not seem to be linear. How do you know that this result still holds in the non-linear case, in particular if the sampling regime is characterized by dominating input noise (which would get nonlinearly transformed)*?

The model in Hermundstad et al. consists of local, nonlinear extraction of features, followed by analysis of how the statistics of these features are represented. The model did not attempt to analyze those nonlinear processes but simply considered their outputs to be “tokens”, which, once extracted, need to be represented and transmitted by central visual areas (Figure 4C of Hermundstad et al.). To apply principles of efficient coding to this process of representation and transmission, we assumed that once the tokens are extracted, the rest of the process is linear and noise is Gaussian. Obviously this is a simplification, but it is one that appears to account for the data. We now summarize this framework in a new paragraph (the third paragraph) in the Introduction.

*How do you justify that more variable features contain more information? In the discrete case, I can see that. However, in the limit of infinitely many images, the multi-point correlations become continuous. In that case I could transform all features by a pointwise monotonic transformation (histogram equalization) that would not change the information content but make all features equally variable*.

As the reviewer indicates, the Hermundstad et al. analysis was predicated on binarizing the continuum of gray levels in natural images, and naturally leads to the question of how the analysis would “scale up” to the full continuum of gray levels. But the notion that histogram equalization would make all features equally variable is not correct, on several grounds. First, although it would equalize first‐order statistics, it would not equalize higher‐order statistics. Second, even for first‐order statistics, histogram equalization is only information‐preserving if one ignores limitations on discriminating nearby gray levels that are arbitrarily close. We know from the work of Chubb et al. that with only 16 gray levels in play (limiting consideration to first‐order statistics), there are only three perceptual coordinates. So something much more interesting than histogram equalization is going on. We’ve begun to look at the three‐gray‐level case with multipoint statistics. Here, there are 66 possible feature dimensions in a 2x2 local neighborhood, and at least a dozen of these are visually perceptible, but not nearly all 66. In any case, this is a complex and interesting matter, and would take us far beyond the scope of this paper.